# Article
# Melon Robotic Grafting: A Study on the Precision Cutting Mechanism and Experimental Validation

Shan Chen [1,2], Huan Liang [3], Qian Zhang [4,*], Qingchun Feng [2], Tao Li [2], Liping Chen [2] and Kai Jiang [2,*]

1   College of Mechanical and Electrical Engineering, Shihezi University, Shihezi 832000, China
2   Research Center of Intelligent Equipment, Beijing Academy of Agriculture and Forestry Sciences, Beijing 100097, China
3   Crops Institute, Wuhan Academy of Agricultural Sciences, Wuhan 430070, China
4   Research Center of Information Technology, Beijing Academy of Agriculture and Forestry Sciences, Beijing 100097, China
*   Correspondence: zhangq@nercita.org.cn (Q.Z.); jiangk@nercita.org.cn (K.J.); Tel.: +86-10-5150-3504 (K.J.)

**Abstract:** The grafting machine cutting step is the core step of the grafting process. The existing grafting machine cutting mechanism adopts fixed angle cutting and manually adjusts the cutting angle based on experience, and the cutting angle is not definite for rootstock cutting in different growth periods. In this paper, we proposed a method to guide the precise cutting mechanism according to internal and external parameters of melon rootstock at a specific period. First, we constructed a cutting model based on internal and external characteristics of rootstock cutting in the growth period of "two leaves and one core" and clarified the safe cutting area. Second, we designed a rotary precision cutting mechanism for rootstock cutting with automatic angle adjustment and constructed the displacement equation of the cutting trajectory of the cutter according to the cutting model. Last, we examined the cutting effect of the precision cutting mechanism and determined the precise cutting angle of the rootstock cutting mechanism in the growth period. Finally, the cutting effect of the precision cutting mechanism was examined, and the precision cutting angle of the rootstock during the growth period was determined. A comparison test between the precision cutting mechanism and the traditional cutting mechanism was carried out, and visual images of the incision were captured and analyzed. The results show that under the five horizontal cutting angles in the safe cutting area of rootstock, the length of the cut surface is inversely proportional to the cutting angle, and the flatness of the cut surface is directly proportional to the cutting angle. Comprehensive evaluation of the length of the cut surface, the cutting success rate, and the quality of cutting revealed that the average cutting angle of the precision cutting mechanism in the safe cutting area (26°) is better than that of the traditional cutting mechanism. It meets the technical requirements of the cutting technology for mechanically grafted rootstocks. The results provide a reference for studying new rootstock precision cutting mechanisms and cutting angle adaptive control models.

**Keywords:** rootstock seedling; medullary cavity model; precision cutting; grafting robot; design; comparative test



## 1. Introduction

Soil pathogens in the right conditions attack the roots or stems of crops, resulting in reduced crop yields or even crop failure. This type of disease is collectively referred to as "soil-borne diseases", and improper fertilization is one of the causes of soil-borne diseases [1,2]. Vegetable grafting technology combines disease-resistant root systems with high-yielding crop varieties to improve the disease resistance and resilience of vegetable crops, significantly reduce dependence on pesticides, reduce pesticide residues in agricultural products, and protect consumer health and food safety [3,4]. Grafting can increase yields by 20–50% and also make vegetables better able to cope with limited environmental conditions and climate change [5]. At the same time, grafting is a horticultural technology

crucial for increasing productivity and faces new challenges and opportunities in China's changing agricultural landscape.

However, manual grafting is still very common. The grafting process needs to follow strict technical standards, but the varying levels of manual skills and the lack of systematic training result in high grafting costs and low efficiency [6–8]. Robotic grafting emerges as a promising solution, automating vegetable seedling grafting with the potential to revolutionize nursery production. Among various techniques, splice grafting is notably favored in machine grafting due to its simplicity and minimal requirements for seedling standardization [9–13]. However, the success of robotic grafting hinges on the precision of rootstock cutting, a core operation within the grafting process. Using the existing grafting machine, the rootstock cutting mechanism adopts a fixed angle to complete the cutting operation of seedlings. As the name suggests, the same cutting parameters are used to complete all the rootstock cutting seedlings, and the angle adjustment process of the cutting mechanism is very cumbersome and unable to obtain a precise cutting angle. Tian et al. selected the range of rootstock and scion cutting angles based on their experience, designed a rotary cutting mechanism, and obtained the optimal parameter combination of cutting radius, cutting speed, and cutting angle [14]. Lu et al. investigated the effects of the sliding cutting angle, cutting edge angle, and average cutting speed on cutting energy consumption and determined the order in which each factor affects the significance of cutting stress [15]. Jiang et al. performed parameter optimization of the rotary cutting mechanism to change the cutting angle by adjusting the spatial position of the center of rotation of the cutter. However, it is a very complex process, and the cutting angle parameter cannot be accurately adjusted solely based on the operator's experience [16,17]. In 2010, ISEKI & CO., Ltd. and Sangyo Research Organization jointly introduced the GRF800-U fully automatic grafting machine in Japan. Rootstock and scion are used in a linear cutting mode, and the cutting parameters are not adjustable [18]. The Helper Robotech Co., Ltd. developed the AFGR-800CS vegetable grafting machine in Korea with linear rootstock cutting and rotary scion cutting, requiring manual adjustment of the cutting angle of both [19]. The ISO Group in the Netherlands introduced the Graft 1200 tomato automatic grafting machine featuring a rootstock and scion integrated synchronized linear cutting mechanism with rootstock and scion flat cutting or 45° cutting to improve the consistency of the cutting angle of the rootstock and the scion. However, the machine did not take into account the differences between the rootstock and the scion seedling stem diameters; therefore, the rootstock and the scion seedling stem diameter consistency requirements are very high [20]. Yang et al. developed the 2JSZ-600 II grafting machine and designed a one-piece bilateral rotary cutting shank, which can synchronize the cutting of rootstock and scion seedlings, but the adjustment of cutting parameters is very cumbersome [21]. The JFT-A1500T grafting machine developed by Hefei AiGraft Robot Technology Co., Ltd. in China, which also adopts a linear cutting method, is designed with a cotyledon positioning mechanism against a piece of cotyledon and the growth point of the rootstock to complete the cutting, which makes it difficult to locate the seedling horizontally and affects the grafting efficiency of the machine [22]. The above scholars used a fixed-angle rotary cutting method or linear cutting. Although the optimal parameters of the cutting mechanism were obtained, they did not consider the variability of seedling morphology and internal medullary cavity characteristics at different growth periods. At present, the facilities of nursery enterprises have limited production conditions and have not yet realized the homogenization and standardization of nursery production. The fixed cutting angle cannot be adapted to the cutting operation of rootstock cutting seedlings of different ages and does not have the adaptability to the same batch of seedlings.

Cut length affects grafted seedling survival. Bausher concluded from tests that the survival of tomato-grafted plants was significantly correlated with the cutting angle (length of cut), with 79%, 81%, and 92% survival for rootstocks with cutting angles of 20°, 45°, and 70°, respectively [23]. Pardo-Alonso et al. investigated the effects of tomato rootstock, scion stem diameter, and cutting angle on grafting survival. These researchers showed

that when the difference between rootstock and scion diameter was small, the larger the cutting angle, the higher the survival rate of grafted seedlings. However, after the cutting angle reached a threshold, the survival rate decreased sharply. Despite the variation in diameter, an increase in grafting angle was associated with the high survival of grafted plants [24,25]. The above studies focused on the cutting angle of tomato seedlings and grafting survival rate. The melon rootstock and scion of the seedling stem cross-section are irregular oval shapes. Generally, the ratio of the length of the long axis of the rootstock and scion fitting surface overlap to the length of the long axis of the rootstock cutting surface is used to express the effective combined area ratio; the larger the fitting area, the higher the grafting survival rate [19]. Liang et al. investigated the effects of three cutting angles (14°, 17°, and 27°) of melon rootstock seedlings on the healing survival and later growth of grafted seedlings. The results showed that the smaller the cutting angle, the earlier the phloem reattachment occurs, and the greater the tensile strength of the scion-rootstock union during the healing period, the more favorable the grafted seedling survival rate [26]. The study did not consider the interaction of cutting angle with cut quality and cutting success, which need to be taken into account in the actual cutting process. A flat cut surface, free of stubble and obstruction, has a greater impact on grafting survival [27].

In summary, we furthered our study by addressing the neglected issue of medullary cutting, a critical factor in grafting success. We constructed the internal medullary cavity model in the period of "two leaves and one core" to determine the safe cutting area. We designed a rootstock precision cutting mechanism, proposed a method of precise control of the cutting angle of melon rootstock cutting based on the rotary cutting method, and compared and analyzed the cutting length, cutting success rate, and cutting quality of the precision cutting mechanism for the five horizontal cutting angles and traditional cutting based on the images. The cutting length, cutting success rate, and cutting quality of the precision cutting mechanism were compared and analyzed to test the precision of the cutting mechanism and determine the precise cutting angle of rootstock cutting in a specific period. The results of the study are expected to lay the foundation for research on the design of precision cutting mechanisms for grafting robotic rootstocks as well as innovative research on vision-driven adaptive cutting methods based on individual seedling information.

## 2. Materials and Methods

### 2.1. Rootstock Cutting Model

Melon rootstock seedlings consist of hypocotyls, cotyledons, and epicotyls with true leaves, as shown in Figure 1. During seed development, the germ and the embryonic axis first break through the seed coat and develop into cotyledons and stems, and the growing point, which consists of the epiblast and the true leaves, grows between the two cotyledons.

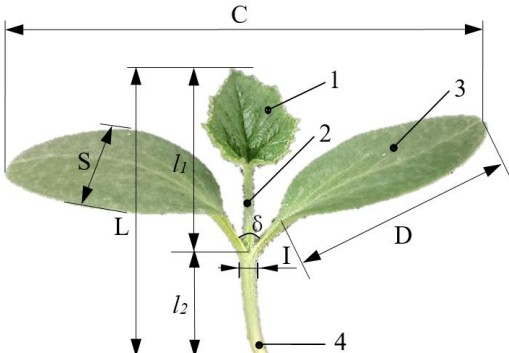

**Figure 1.** Melon rootstock seedlings. Components: 1. True leaf. 2. Epiblast. 3. Cotyledon. 4. Hypocotyl 1 + 2. Growing point. C is cotyledon span, mm. S is cotyledon width, mm. D is cotyledon length, mm. I is seedling stem short axis, mm. $l_1$ is growth point height, mm. $l_2$ is hypocotyl length, mm. L is plant height, mm. δ is the petiole angle.

The rootstock grows to a certain stage where a medullary cavity will appear inside the stem. If the cutting angle is not precise, it is easy to cut through the medullary cavity, which will lead to grafting failure. Avoiding penetration of the medullary cavity is essential, and the cutting area needs to be determined according to the growth characteristics of the medullary cavity inside the seedling stem to ensure that a cotyledon and the growing point are completely removed without penetrating the medullary cavity. The requirement that the splice grafting method of cutting be able to remove exactly one cotyledon is easy to achieve in terms of the appearance of the seedling. However, there is no way of knowing whether or not the medullary cavity has been cut through unless each seedling is dissected and cut according to the distribution of the medullary apexes, which is impossible to achieve in the actual grafting process. Therefore, it is necessary to determine the extent of cutting by combining the appearance of the seedling with the distribution of internal medullary cavity characteristics and growth dynamics.

The appearance of the rootstock seedling is shown in Figure 2a. The image is frontal to the direction of the unfolding of the two cotyledons, with the bases of the two cotyledons intersecting at one point, i.e., the external two-cotyledon intersection point $G$. The medullary cavity structure is noted along the short axis (two cotyledons unfolding direction) of the central axis of the top-down dissection to expose the medullary cavity structure, such as shown in Figure 2b, marking the medullary cavity apex $O$, the growth point of the left and right basal points of $A_1$ and $A_2$, the intersection of the two cotyledons $G$, and other target features, and the triangle formed by the line connecting the above feature points is the limit of the cutting area.

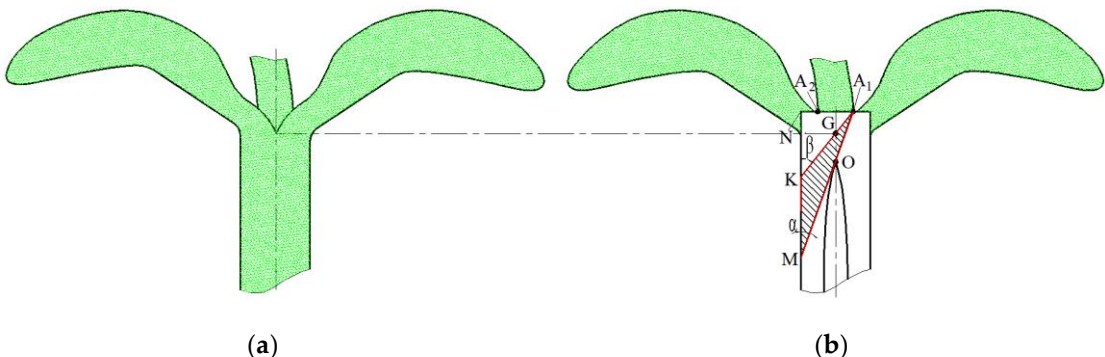

|          |          |
|:--------:|:--------:|
| (**a**)  | (**b**)  |

**Figure 2.** Precision cutting model construction of rootstock. (**a**) External morphology; (**b**) Model of pith cavity structure.

The maximum cutting angle $\beta$ is as follows:

$$\beta = \arctan\frac{l_{A_1 A_2}}{2l_{OA}} \tag{1}$$

The minimum cutting angle $\alpha$ is as follows:

$$\alpha = \arctan\frac{l_{A_1 A_2}}{2(l_{OA} - l_{OG})} \tag{2}$$

The maximum cut length $l_{A_1 M}$ is as follows:

$$l_{A_1 M} = \frac{l_{A_1 N}}{\sin \alpha} \tag{3}$$

The minimum cut length $l_{A_1 K}$ is as follows:

$$l_{A_1 K} = \frac{l_{A_1 N}}{\sin \beta} \tag{4}$$

where $l_{A_1A_2}$ is the growth point width, mm; $l_{OA}$ is the vertical distance from the apex of the medullary cavity to the two growth points, mm; $l_{OG}$ is the distance from the apex of the medullary cavity to the intersection of the two cotyledons, mm; and $l_{A_1N}$ is the distance from the right base of the growth point to the left edge of the stem, mm.

In Figure 2, $A_1$ is the right base of the growth point; $A_2$ is the left base point of the growth point; $N$ is the vertex of the left edge of the stem; $G$ is the outer two cotyledon intersection; $O$ is the apex of the medullary cavity; $K$ is the intersection of the extension of $A_1G$ with the left edge of the stem; $M$ is the intersection of the extension of $A_1O$ with the left edge of the stem; $\alpha$ is the angle between $A_1M$ and the vertical direction of the stem (minimum limiting cutting angle); $\beta$ is the angle between $A_1K$ and the vertical direction of the stem (maximum limit cutting angle); and $\Delta A_1KM$ is the limit cutting area.

### 2.2. Precision Cutting Mechanism for Rootstocks

### 2.2.1. Structural Composition and Working Principle

The rootstock cutting mechanism is the core component of the grafting machine, completing the rootstock seedling cotyledon and growth point cutting. The grafting robot workflow involves the following steps (Figure 3): ① The system is powered on, and the gas source is connected. The rootstock and scion clamping and carrying mechanisms are reset to the seedling-loading station. ② At the seedling-loading station, the seedling-loading operation of the rootstock and scion is completed by two persons, and the clamping claw clamps and picks up the seedling of the rootstock and scion. Then, the rootstock and scion-clamping and carrying mechanisms rotate 90° relative to each other to carry the seedling to the cutting station. ③ At the cutting station, the cutting mechanism cuts the rootstock and scion, separately, and the rootstock and scion clamping and carrying mechanisms continue to rotate 90° relative to each other to reach the buttressing station. ④ At the buttressing station, the two clamping claws are extended, and the cutting of the rootstock and scion is buttressed to fit. The automatic clip-feeding mechanism outputs grafting clips to clamp and fix the rootstock and scion to complete a grafted seedling. Finally, the clamping claw, cutting, and clip-feeding mechanisms are reset in turn.

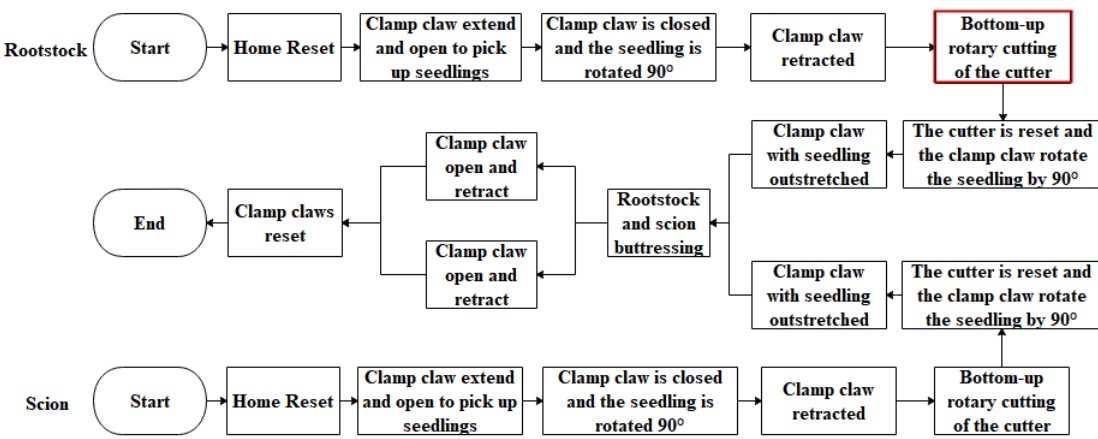

**Figure 3.** Grafting robot workflow. The step marked with red framed indicates the position in the whole grafting process and also the focus of this paper.

The Rootstock precision cutting mechanism consists of a frame, fixed plate, rotary cylinder, connected plate, cutter arm, cutter seat, cutter, support baffle, pressure seedling piece, pressure seedling cylinder, adjustable rack, and *XY* motorized sliding table (stepping motor, guide rail, sliding table, photoelectric sensors), as shown in Figure 4. The seedling pressure cylinder is mounted on one side of the frame at a certain inclination through the adjustable rack, and the seedling pressure piece is mounted on the front end of the seedling pressure cylinder. The seedling pressure cylinder drives the seedling pressure piece to move down the cotyledons of the rootstock that are to be retained so that the cotyledon

to be excised is completely exposed to the growth point. The rotary cylinder is connected and mounted on the other side of the bracket in sequence with the connected plate, *XY* motorized sliding table, and fixed plate. The cutter is mounted on the output shaft of the rotary cylinder through the cutter seat and the cutter arm, and the rotary cylinder can drive the cutter to rotate rapidly by 180° from the bottom to the top or top to the bottom to complete the operation of a cotyledon and growth point removal. An *XY* motorized sliding table is used to complete the position adjustment of the vertical and horizontal directions of the rotating center of the cutter. The mechanism allows precise control of the cutting angle using a rootstock cutting model and a control system.

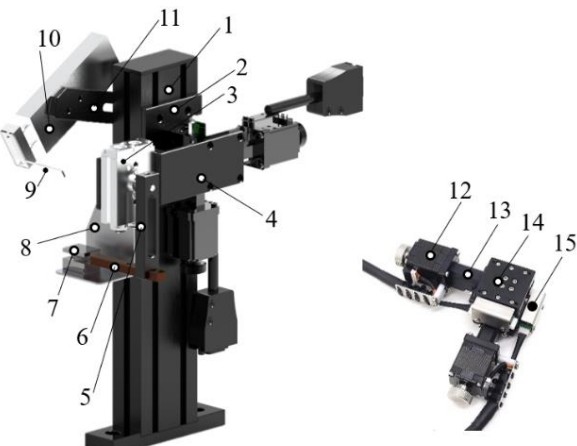

**Figure 4.** Rootstock precision cutting mechanism. 1. Frame; 2. Fixed plate; 3. Rotary cylinder; 4. Connected plate; 5. Cutter arm; 6. Cutter seat; 7. Cutter; 8. Support baffle; 9. Pressure seedling piece; 10. Pressure seedling cylinder; 11. Adjustable rack; 12. Stepping motor; 13. Guide rail; 14. Sliding table; 15. Photoelectric sensors.

Work process: ① First, the cutting angle parameters are determined according to the rootstock cutting model, and the center of the cutter is moved to the predetermined position by the control system driving the *XY* motorized sliding table. ② Start the grafting machine. After the seedling-loading operation is performed by two people, the clamping and carrying mechanism will clamp the rootstock and carry it to the cutting station. The pressure seedling cylinder will connect the positive pressure air source to drive the pressure seedling piece to extend and press down the rootstock's reserved cotyledon. ③ The rotary cylinder is connected to a positive pressure air source to drive the cutter to rotate rapidly to remove cotyledon and the growth point of the rootstock waiting to be cut. ④ The clamping and carrying mechanism carries the finished rootstock cutting to the buttressing station, while all the cylinders of the cutting mechanism cut off the positive pressure air source to complete the reset.

### 2.2.2. Cutting Angle Control System

To realize the precise control of the cutting angle of rootstock cutting seedlings, a precise control system based on a motorized slide table was developed. The system consists of a stepper motor controller, *XY* motorized slide table, driver, switching power supply, etc., as shown in Figure 5. The motorized slide table is type FEX4015-LBNL with the following features: stroke ± 7.5 mm, ball screw lead 6 mm, and accuracy ± 0.1 mm. There are 3 photoelectric sensors inside the slide table, which are used to detect the 3 positions of the slide origin and the upper and lower limits. The controller and driver models are HJ40 (L)20 and DM3622, respectively. Based on the rootstock cutting model, the precise cutting angle and cut length information were determined, and the amount of change in the spatial displacement of the cutter center was calculated. The *XY* motorized slide table control program is written in the controller's operation interface, which can directly input the

moving distance in both positive and negative directions to realize the precise regulation of the cutting angle.

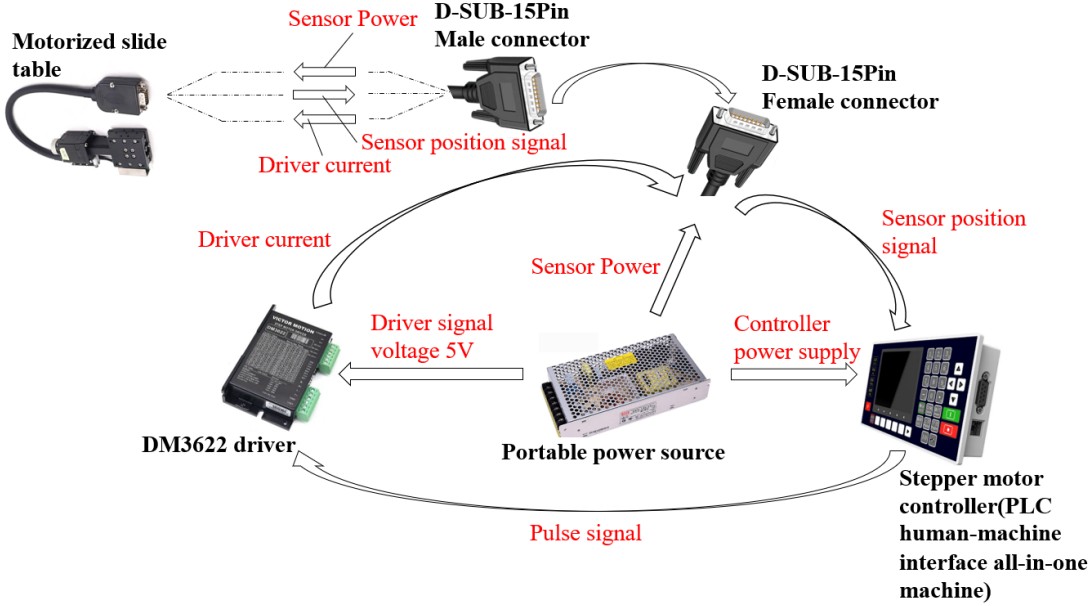

**Figure 5.** Rootstock cutting angle control system.

The control system workflow is shown in Figure 6. First, the system parameters are initialized, and the spatial position of the center of the cutter corresponding to the minimum limit cutting angle of the rootstock at a certain seedling age is set as the initial position of the cutting mechanism. Based on the precise cutting angle determined by the rootstock cutting model, the corresponding coordinate change data for the cutter center position were calculated. The control program is written in the controller, and the motorized sliding table drives the cutter to move to the set cutting angle, completing the automatic control of the cutting angle.

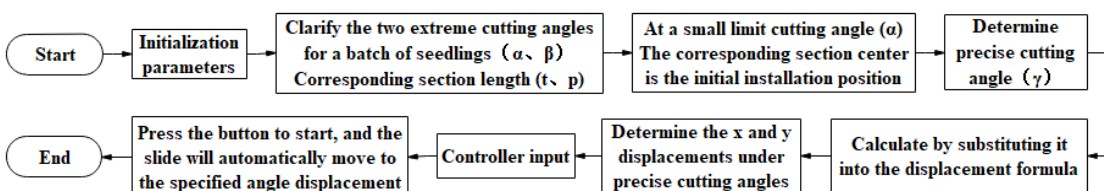

**Figure 6.** Cutting angle control process.

### 2.2.3. Selection of Rootstock Cutting Direction

Choosing the proper cutting direction can improve the quality of seedling cutting. The rootstock cutting direction was analyzed as shown in Figure 7. The bottom-up rotary cutting method was chosen for rootstock cutting for the following reasons: ① It avoids pulling the epidermis of the stem. When the cutter cuts from the bottom up, it cuts the hypocotyl epidermis, phloem, formative layer, and growth point sequentially, which is conducive to the formation of a good incision. On the contrary, if cutting is performed from the top down, the cutter first passes through the growth point and finally passes through the epidermis of the hypocotyl, which easily causes epidermal pulling, and the epidermal damage site is prone to disease infection, affecting the survival of grafted seedlings. ② The direction keeps the cutting process stable. When the cutter cuts from the bottom up, the rootstock seedling is fixed in the clamping claw, and the stem and cotyledons are positioned on the side opposite to the cutting direction using the top bar and the pressure seedling piece to increase the stability of the rootstock cutting process. ③ The cutting direction can

provide better vision and maneuverability. Bottom-up cutting provides better visibility and maneuverability, allowing the operator to clearly observe the cut and facilitating more accurate control of the depth and angle of the cut.

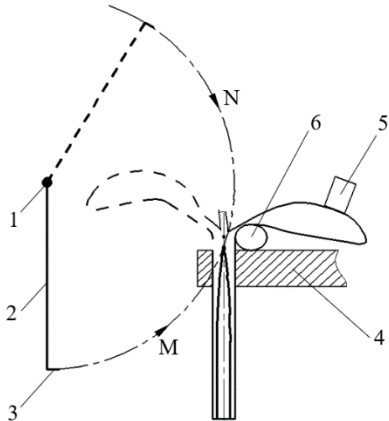

**Figure 7.** Diagram of rootstock cutting direction. 1. Cutter rotation center; 2. Cutter arm; 3. Cutter; 4. Clamping claw; 5. Pressure seedling piece; 6. Imitation support rack. M is the bottom-up cutting trajectory of the cutter, and N is the top-down cutting trajectory of the cutter.

### 2.3. Cutting Angle Adjustment Model

In Section 2.1, the rootstock pith cavity model and the limit cutting area are described in detail. To realize the automatic control of the cutting angle of the rootstock, it is necessary to establish the relationship between the cutting angle and the change of the position of the center of the cutter (the cutting angle adjustment function), i.e., the change of the cutting angle is transformed into the change of the displacement of the center of the cutter. When the cutting mechanism inputs any cutting angle, the center position of the cutter is adjusted by the rapid movement of the *XY* motorized slide table. To determine the adjustment process of the cutting angle, a geometric model of the angular change of the cutting mechanism is established to clarify the range of displacement change of the center of the cutter in the *X-Y* direction between the two limited cutting angles and to determine the amount of displacement change of the center of the cutter corresponding to any cutting angle in the limit cutting area.

#### 2.3.1. Rotary Cutting Regulation Process

The rotary cutting process of the rootstock is shown in Figure 8a. When the center of rotation of the cutter is located, the cutting trajectory passes through the maximum limit cutting angle, i.e., through the intersection of the two cotyledons and the right base of the growth point. When the center of rotation of the cutter is located, the cutting trajectory passes through the minimum limiting cutting angle, i.e., through the apex of the medullary cavity and the right base of the growth point. Take the horizontal direction of the *X*-axis and the vertical direction of the *Y*-axis, and establish the plane right-angle coordinate system *XOY*, as shown in Figure 8b.

In the figure:

$B_2W$—Auxiliary line 1, which is parallel to the *X*-axis intersecting the *MN* extension line, denoted by *a*, mm;
$WQ$—Auxiliary line 2, which is located on the extension line of *MN* and parallel to the *Y*-axis, denoted by *b*, mm;
$A_1E$—Cutting length of rootstock at a large limit cutting angle, denoted by *t*, mm;
$A_1M$—Cutting length of rootstock at a small limit cutting angle, denoted by *p*, mm;
$B_1B_2$—Actual displacement of the cutter center, denoted by *c*, mm;
$B_1B_3$—Sub-displacement of the cutter center from the origin *Q* in the vertical direction, denoted in *y*, mm;

$B_2B_3$—Sub-displacement of the cutter center from the origin $Q$ in the horizontal direction, denoted in $x$, mm.

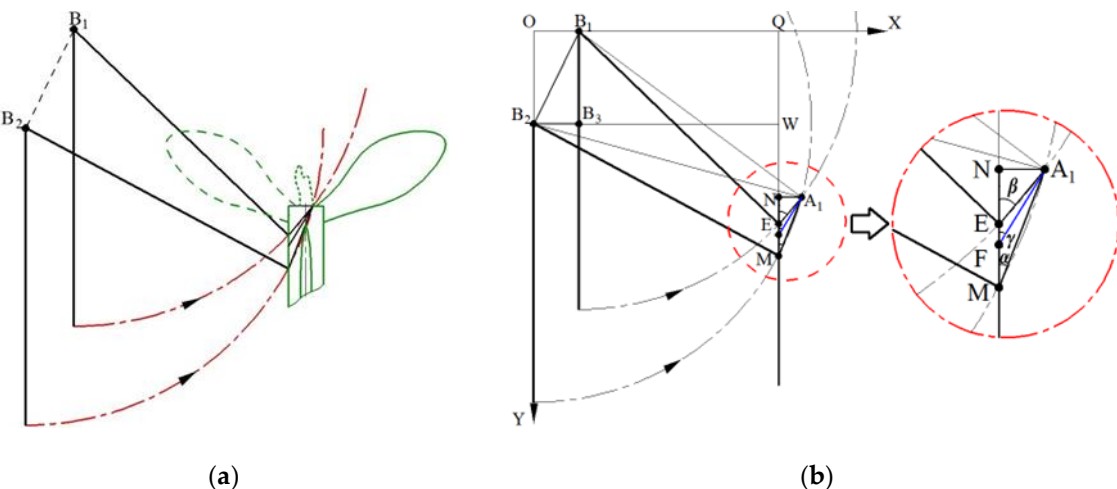

(**a**)                                                                (**b**)

**Figure 8.** Rootstock cutting angle adjustment methods. (**a**) Schematic diagram of the cutting process; (**b**) Geometric modeling of the cutting process. $A_1F$ is the cutting length of rootstock at an average cutting angle in the limit cutting area; $\gamma$ is the average cutting angle. They will be involved later.

In $\Delta B_2 A_1 M$, the following equations can be obtained:

$$\angle B_2 M A_1 = \cos^{-1}\left(\frac{p}{2r}\right) \tag{5}$$

In $\Delta B_1 A_1 E$, the following equations can be obtained:

$$\angle B_1 E A_1 = \cos^{-1}\left(\frac{t}{2r}\right) \tag{6}$$

$$\begin{cases} \angle B_2 M N = \cos^{-1}\left(\frac{p}{2r}\right) - \alpha \\ \angle B_1 E N = \cos^{-1}\left(\frac{t}{2r}\right) - \beta \end{cases} \tag{7}$$

$$\begin{cases} A = r\sin\left[\cos^{-1}\left(\frac{p}{2r}\right) - \alpha\right] \\ b = r\sin\left[\cos^{-1}\left(\frac{t}{2r}\right) - \beta\right] \end{cases} \tag{8}$$

where

$$\begin{cases} \angle E A_1 M = \beta - \alpha \\ \angle B_2 A_1 M = \angle B_2 M A_1 \\ \angle B_2 A_1 E = \angle B_2 A_1 M - \angle E A_1 M \\ \angle B_1 A_1 E = \angle B_1 E A_1 \\ \angle B_1 A_1 B_2 = \angle B_1 A_1 E - \angle B_2 A_1 E \end{cases} \tag{9}$$

The following can be concluded:

$$\angle B_1 A_1 B_2 = \cos^{-1}\left(\frac{t}{2r}\right) - \cos^{-1}\left(\frac{p}{2r}\right) + \beta - \alpha \tag{10}$$

In addition,

$$\begin{cases} c^2 = 2r^2 - 2r^2\cos\left[\cos^{-1}\left(\frac{t}{2r}\right) - \cos^{-1}\left(\frac{p}{2r}\right) + \beta - \alpha\right] \\ x = a - b \end{cases} \tag{11}$$

The cutter center displacement equation can be obtained as follows:

$$\begin{cases} x = r\sin\left[\cos^{-1}\left(\frac{p}{2r}\right) - \alpha\right] - r\sin\left[\cos^{-1}\left(\frac{t}{2r}\right) - \beta\right] \\ y = \sqrt{2r^2 - 2r^2\cos\left[\cos^{-1}\left(\frac{t}{2r}\right) - \cos^{-1}\left(\frac{p}{2r}\right) + \beta - \alpha\right] - r^2\sin\left[\cos^{-1}\left(\frac{p}{2r}\right) - \alpha\right]^2 - r^2\sin\left[\cos^{-1}\left(\frac{t}{2r}\right) - \beta\right]^2} \end{cases} \tag{12}$$

### 2.3.2. Determination of the Initial Value of the Cutting Angle

The spatial position of the center of the cutter is determined using the minimum limit cutting angle as the initial value of the rootstock cutting mechanism. The center of the cutter is the origin $O$. Set the horizontal mounting position as $X$ (horizontal distance from the center of the cutter to the center of the seedling clamping hand) and the vertical mounting position as $Y$ (vertical distance from the center of the cutter to the platform of the grafting machine). When the cutting angle increases, the position of the center of the cutter changes in the process of the horizontal direction to the right and vertical direction up.

Initial horizontal mounting distance of the cutter center:

$$X = a + (p\,\sin\alpha)/2 \tag{13}$$

Initial vertical mounting distance from the center of the cutter:

$$Y = \sqrt{r^2 - a^2} - \frac{p\sin\alpha}{2} + 193 \tag{14}$$

The process of changing the cutter center position is represented by the following equations:

$$\begin{cases} X = r\sin\left[\cos^{-1}\left(\frac{p}{2r}\right) - \alpha\right] + \frac{p\sin\alpha}{2} \\ Y = \sqrt{r^2 - r^2\sin\left[\cos^{-1}\left(\frac{p}{2r}\right) - \alpha\right]^2} - \frac{p\cos\alpha}{2} + 193 \end{cases} \tag{15}$$

### 2.4. Tests

#### 2.4.1. Test Content

Seedling process: Jingxin rootstock No. 2 white seeded pumpkin seeds were selected, hole tray size (5 × 10), and the number of seedlings was 500. Vermiculite, peat, and perlite were evenly mixed and stirred in a ratio of 1:1:1. The substrate was watered well and loaded into cavity trays and pressed into holes, which were 20 mm in length, 20 mm in width, and 10 mm in depth. Place ungerminated seeds in the holes of the hole trays, and cover them with substrate and water thoroughly. The hole trays were placed in sunny greenhouse nursery beds of the Beijing Academy of Agricultural and Forestry Sciences, with a daytime temperature of 28 °C, a nighttime temperature of 25 °C, and a humidity of 60–70%. Rootstocks were cultivated until day 5 after both cotyledons were fully expanded, as shown in Figure 9.

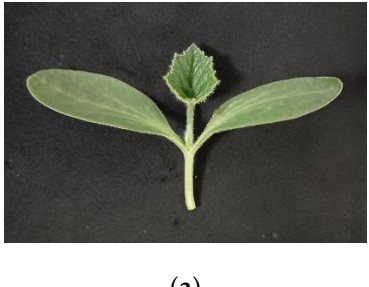

(**a**)

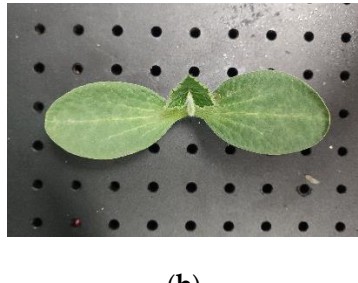

(**b**)

**Figure 9.** Rootstock seedlings. (**a**) Front view; (**b**) vertical view.

The following tests were performed: measurement of internal and external characteristic parameters of rootstock seedlings, precision cutting tests, and comparative tests on cutting quality. Details are as follows:

Test 1: Measurement of internal-external characterization parameters of rootstocks

We used digital vernier calipers (range: 0–150 mm, index value: 0.02 mm, manufacturer: Shida Tools (Shanghai, China) Co., Ltd.) to measure 30 rootstock seedlings' hypocotyl length, plant height, growth point height, seedling stem short axis, seedling stem long axis, cotyledon length, cotyledon width, cotyledon span, and other parameters. The internal pith cavity characteristics of rootstocks are difficult to measure directly. We used a CCD (Charge coupled Device) camera to look at the internal pith cavity structure of the rootstocks that had been cut open. We measured parameters, such as the distance between the growth point left base point and the right edge of the seedling stem, the width of the growth point, the distance between two growth points and the pith cavity apex, the petiole pinch angle, and the distance between the pith cavity apex and the cotyledon intersection. Substitute Equations (1)–(4) to calculate the two limiting cutting angles and the length of the cut.

Test 2: Precision cutting test

To validate the cutting performance of the rootstock precision cutting mechanism, the most suitable cutting parameters were determined. Five horizontal cutting angles were taken in the limited cutting area of the rootstocks, and 50 rootstocks were cut in each group of cutting angles to examine the quality of the cut surface, cutting effect, and success rate. Cutting Angle Adjustment Process: Adjust the center of the cutter to the initial position using the minimum limit cutting angle $\alpha$ as the initial cutting angle. Each group of cutting angles is adjusted on this basis, and the five horizontal cutting angles are separately used as the target cutting angle $\beta$. Equation (12) calculates the displacement of the cutter's center and inputs it into the control system to complete the cutting angle adjustment.

Test 3: Cutting quality comparison test

A comparative test of the cutting performance of the conventional cutting mechanism (2TJGQ-800 grafting machine) and the precision cutting mechanism (five horizontal cutting tests in Test 2) was performed, and the evaluation indexes included the length of the cut surface, the quality of the cut surface, and the cutting success rate. Conventional cutting mechanisms require manual adjustment of the cutting angle, and the actual cutting angle data are not available. Images of the cut surface of the rootstock seedlings were captured using the visual image testing system to analyze whether the medullary cavity was cut through, whether the cotyledons were cut, whether the growth points were removed cleanly, whether the leaf crowns were removed, etc., and the occurrence of any of the above situations was recognized as a cutting failure. The length of the cut was measured using digital vernier calipers, and the success rate of each set of cutting tests was recorded in statistics.

2.4.2. Test Apparatus

The rootstock precision cutting test bench includes the upper seedling table, clamping and handling mechanism, cutting mechanism, and control system, etc., as shown in Figure 10.

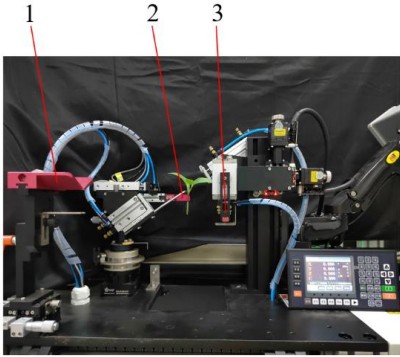

**Figure 10.** Rootstock precision cutting test bench. 1. Upper seedling table; 2. Clamping and carrying mechanism; 3. Cutting mechanism.

Experimental process:

① Manually adjust the direction of the two cotyledons of the rootstock and the height of the upper seedling, and place the rootstock seedling onto the upper seedling table. ② The rootstock clamping and carrying mechanism completes the rootstock clamping and rotates 90° anticlockwise to transport the rootstock to the cutting station. ③ To fully expose the growth point and another cotyledon, the pressure seedling cylinder moves the seedling pressure piece to extend and press down on the retained piece of cotyledon. The rotary cylinder then moves the cutter quickly from bottom to top, removing the growth point and cotyledon. ④ Cutting angle adjustment. Use the rootstock cutting model to calculate the cutting angle and cutter center displacement change data, and complete the data input and regulation operation in the control system.

The visual image test system consists of a CCD camera (model MV-EM120C), lens (model AFT-ZML1000), bracket, LED fill light, laptop, white backdrop, cutter, and dongle, as shown in Figure 11.

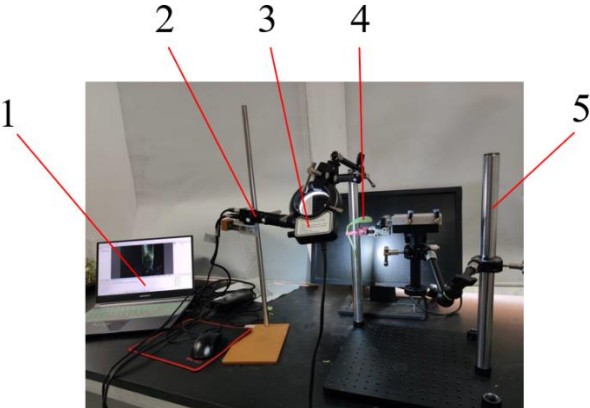

**Figure 11.** Visual Image Test System. 1. Laptop; 2. CCD camera; 3. LED fill light; 4. Seedling after cutting; 5. Bracket.

*2.5. Data Processing*

A one-way ANOVA (analysis of variance) with multiple post-hoc comparisons analysis of the LSD (Least Significant Difference) test was used for statistical assessment ($p < 0.05$) using SPSS 26 and visualized using Origin 2018.

**3. Results**

*3.1. Parameters of Internal-External Characterization of Rootstocks*

The results of the statistics of the parameters of the appearance characteristics of the rootstock seedlings are presented in Table 1.

Statistical results of the parameters characterizing the internal medullary cavity of the rootstock are shown in Table 2. Substituting the relevant characteristic parameters into Equations (1)–(4), a maximum limit cutting angle of 32.53° and a length of the cut surface of 4.67 mm, a minimum limit cutting angle of 19.19° and a length of the cut surface of 7.64 mm, and an average cutting angle of 25.86° and an average cut length of 6.15 mm were obtained. If the two limit cutting angles are chosen as the precise cutting angles, situations such as cutting through the medullary cavity or unclean removals of cotyledons may occur. Therefore, safer cutting angles need to be selected within the rootstock-limit cutting area.

**Table 1.** Characteristic parameters of rootstock appearance.

| Seedlings | Hypocotyl Length/(mm) | Plant Height/(mm) | Growth Point Height/(mm) | Seedling Stem Short Axis/(mm) | Seedling Stem Long Axis/(mm) | Cotyledon Length/(mm) | Cotyledon Width/(mm) | Cotyledon Span/(mm) | Petioles Angle/(°) |
|---|---|---|---|---|---|---|---|---|---|
| 30 | 28.96 ± 3.95 | 73.30 ± 4.74 | 44.98 ± 4.55 | 3.02 ± 0.12 | 3.46 ± 0.16 | 53.51 ± 2.18 | 33.13 ± 2.12 | 101.34 ± 5.86 | 65.88 ± 3.87 |

**Table 2.** Characteristic parameters of internal medullary cavity of rootstocks.

| Seedlings | Left Base of Growth Point—Right Edge of Seedling Stem/(mm) | Width of Growth Point/ (mm) | Two Growth Points—Medullary Apex/(mm) | Medullary Apex-Cotyledonary Intersection/(mm) | Maximum Cutting Angle $\beta$ /(°) | Minimum Cutting Angle $\alpha$ /(°) | Minimum Cut Length $p$ /(mm) | Maximum Cut Length $t$ /(mm) | Average Cutting Angle $\gamma$ /(°) | Average Cut Length/(mm) |
|---|---|---|---|---|---|---|---|---|---|---|
| 30 | 2.51 | 2.13 | 3.06 | 1.39 | 32.53 | 19.19 | 7.64 | 4.67 | 25.86 | 6.15 |

The average value of the rootstock limit cutting area was taken as a base, and values were taken at equal intervals of 2° for both sides to form five levels of cutting angles, and the length of the cut, quality of cut, and cutting success rate were explored for the five levels of the cutting angle. The average cutting angle of the rootstock was 25.86°, which was rounded to 26°, and the five levels cutting angles were 22°, 24°, 26°, 28°, and 30°. The minimum limit cutting angle of 19.19° was used as the initial angle $\alpha$, and five levels of the cutting angle were used as the target angle $\beta$ for each batch of cutting adjustment. The rotating arm r of the cutter is known to be 68 mm, and the distance of movement from the initial installation position of the cutter center in the vertical direction $Y$ and in the horizontal direction $X$ needed for each target angle is calculated using Equation (12), and results are shown in Table 3. According to Equation (15), the initial installation position of the cutter centers $X$ and $Y$ are 64.12 mm and 283.85 mm, respectively.

**Table 3.** Changes in the cutter center position for different cutting angles.

| Num. | Initial Angle $\alpha$/(°) | Target Angle $\beta$/(°) | Initial Angular Cut Length $p$ /(mm) | Target Angle Cut Length $t$ /(mm) | Cutter Center Horizontal Direction Displacement $x$/(mm) | Cutter Center Vertical Direction Displacement $y$/(mm) |
|---|---|---|---|---|---|---|
| 1 | 19.19 | 22 | 7.64 | 6.70 | 0.152 | 0.447 |
| 2 | 19.19 | 24 | 7.64 | 6.17 | 1.086 | 3.038 |
| 3 | 19.19 | 26 | 7.64 | 5.73 | 2.102 | 5.543 |
| 4 | 19.19 | 28 | 7.64 | 5.35 | 3.197 | 7.971 |
| 5 | 19.19 | 30 | 7.64 | 5.02 | 4.369 | 10.328 |

*3.2. Effect of Different Cutting Angles on the Length of the Cut Surface*

The results of the descriptive statistics of the five levels of the cutting angle on the length of the cuts are shown in Table 4. The mean, standard deviation, standard error, 95% confidence intervals, and maximum and minimum values of the dependent variable cut length are given in Table 4.

**Table 4.** Descriptive statistics for cuts length.

| Cutting Angle | Minimum Cut Length/(mm) | Maximum Cut Length/(mm) | Mean Value of Cut Length/(mm) | Standard Error | Standard Deviation | 95% of Mean Confidence Interval | |
|---|---|---|---|---|---|---|---|
| | | | | | | Lower Limit | Upper Limit |
| 22° | 6.25 | 6.74 | 6.52 | 0.02 | 0.12 | 6.48 | 6.57 |
| 24° | 5.78 | 6.38 | 6.03 | 0.03 | 0.14 | 5.99 | 6.09 |
| 26° | 5.34 | 6.01 | 5.63 | 0.04 | 0.21 | 5.55 | 5.70 |
| 28° | 4.88 | 5.64 | 5.27 | 0.04 | 0.23 | 5.19 | 5.36 |
| 30° | 4.65 | 5.14 | 4.96 | 0.02 | 0.13 | 4.91 | 5.01 |

The effect of the five levels of the cutting angle versus conventional cutting angles on the length of the cut is shown in Figure 12. Figure 12a shows the cutting lengths corresponding to the five levels of the cutting angle; the larger the cutting angle, the smaller the cutting length. Figure 12b shows the data comparing the length of the cut surface at an average cutting angle of 26° with the conventional cutting angle, and the values obtained do not significantly vary from each other.

To further analyze the effect of the cutting angle on the length of the cut surface, the hypothesis test was first conducted, and the hypothesis of the ANOVA chi-square test was set such that there is no significant difference in the length of the cut surface for the five levels of the cutting angle. The results of the one-way ANOVA with five levels of cutting angles are shown in Table 5. The between-group sum of squares was 45.75, with a degree of freedom df of 4 and a mean square of 11.437. The within-group sum of squares was 4.262, with a degree of freedom df of 145 and a mean square of 0.029. The F-statistic was 389.13. The hypothesis should be rejected since the probability of

concomitance Sig (*p*-value) = 0.000 < 0.05 for intergroup comparisons indicates that the five levels of cutting angles are significantly different for the length of the cut surface.

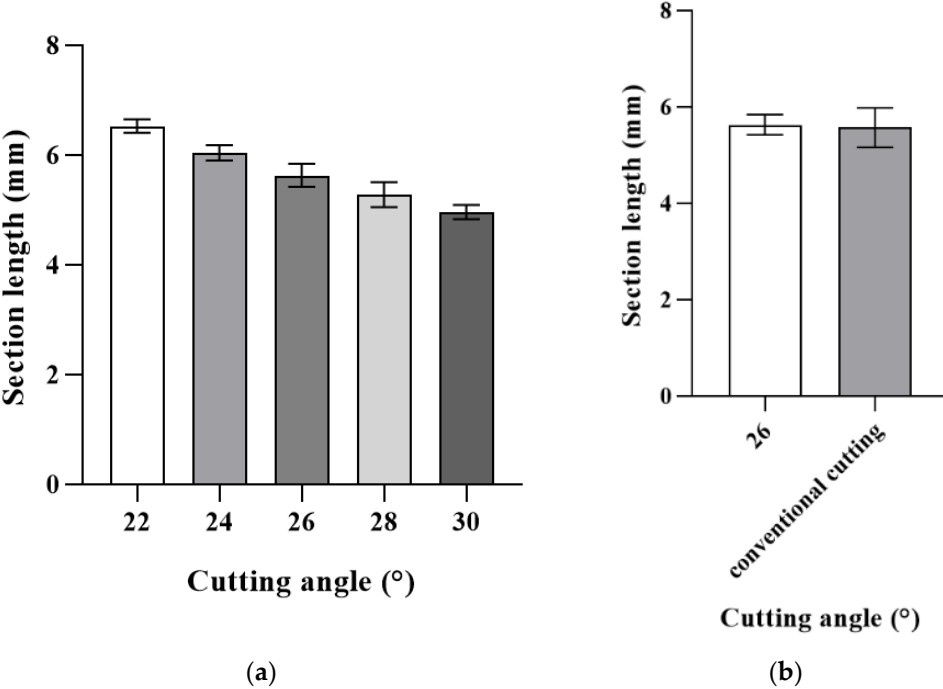

|(**a**)|(**b**)|

**Figure 12.** Effect of cutting angle on cut length. (**a**) Length of the cut at five levels of cutting angle; (**b**) Comparison of the cut length for a cutting angle of 26° and the conventional cutting angle.

**Table 5.** One-way ANOVA.

| Source of Variation | Sum of Squares | df | Mean Square | F-Value | *p*-Value |
|---|---|---|---|---|---|
| Intergroup | 45.750 | 4 | 11.437 | 389.13 | <0.000 |
| Intragroup | 4.262 | 145 | 0.029 | | |
| Aggregate | 50.012 | 149 | | | |

ANOVA and multiple comparisons of cut length results are shown in Table 6. For separate cutting angle two-by-two comparisons, for example, cutting angle 22° compared with 24°, 26°, 28°, and 30°, mean differences of 0.49, 0.90, 1.25, and 1.56 were obtained, respectively. For all of the comparisons, probability Sig. = 0.00 < 0.05, which indicates that 22° and the other four cutting angles are significantly different. In addition, the mean difference in the length of the facet length is significantly lower than the other four cutting angles, which indicates that the length of the cut at the cutting angle of 22° is longer than the length of the cut at the other four angles. Similarly, the cut lengths at the 5 cutting angles are ranked as 22° > 24° > 26° > 28° > 30°.

The statistical results of the five levels of the cutting angles corresponding to the length of the cut are shown in Table 7. The measured value of the length of the cut surface is smaller than the theoretical value, and the differences between the two are 0.18, 0.13, 0.1, 0.07, and 0.06, respectively. As the cutting angle increases, the measured value of the length of the cut surface is close to the theoretical value. The degree of concavity of the cut surface is the largest at the cutting angle of 22°. These results indicate that the larger the cutting angle is, the flatter the cut surface is.

**Table 6.** Multiple comparisons of cut length results (LSD).

| (I) Cutting Angle/(°) | (J) Cutting Angle/(°) | Mean Difference (I-J)/(°) | Standard Error | Significance | 95% of Mean Confidence Interval | |
|---|---|---|---|---|---|---|
| | | | | | Lower Limit | Upper Limit |
| 22 | 24 | 0.49 * | 0.04 | 0.00 | 0.40 | 0.57 |
| | 26 | 0.90 * | 0.04 | 0.00 | 0.81 | 0.99 |
| | 28 | 1.25 * | 0.04 | 0.00 | 1.16 | 1.34 |
| | 30 | 1.56 * | 0.04 | 0.00 | 1.48 | 1.65 |
| 24 | 22 | −0.49 * | 0.04 | 0.00 | −0.57 | −0.40 |
| | 26 | 0.41 * | 0.04 | 0.00 | 0.33 | 0.50 |
| | 28 | 0.76 * | 0.04 | 0.00 | 0.68 | 0.85 |
| | 30 | 1.08 * | 0.04 | 0.00 | 0.99 | 1.17 |
| 26 | 22 | −0.90 * | 0.04 | 0.00 | −0.99 | −0.81 |
| | 24 | −0.41 * | 0.04 | 0.00 | −0.50 | −0.33 |
| | 28 | 0.35 * | 0.04 | 0.00 | 0.26 | 0.44 |
| | 30 | 0.67 * | 0.04 | 0.00 | 0.58 | 0.75 |
| 28 | 22 | −1.25 * | 0.04 | 0.00 | −1.34 | −1.16 |
| | 24 | −0.76 * | 0.04 | 0.00 | −0.85 | −0.68 |
| | 26 | −0.35 * | 0.04 | 0.00 | −0.44 | −0.26 |
| | 30 | 0.31 * | 0.04 | 0.00 | 0.23 | 0.40 |
| 30 | 22 | −1.56 * | 0.04 | 0.00 | −1.65 | −1.48 |
| | 24 | −1.08 * | 0.04 | 0.00 | −1.17 | −0.99 |
| | 26 | −0.67 * | 0.04 | 0.00 | −0.75 | −0.58 |
| | 28 | −0.31 * | 0.04 | 0.00 | −0.40 | −0.23 |

"*" Indicates a significant difference between different cutting angles.

**Table 7.** Corresponding cut lengths for the five levels of cutting angles.

| | 22° | 24° | 26° | 28° | 30° |
|---|---|---|---|---|---|
| Theoretical value of cut length (mm) | 6.70 | 6.17 | 5.73 | 5.35 | 5.02 |
| Measured value of cut length (mm) | 6.52 | 6.04 | 5.63 | 5.28 | 4.96 |

Rootstock and scion cuts are butt-jointed, and grafting clips are clamped in place as shown in Figure 13. Using a known grafting clip height of 10 mm to the rootstock and scion cutting close to the butt, the scion does not appear to collapse in principle. When the vertical height of the cut surface is greater than 8 mm, the scion easily collapses. When the vertical height of the cut surface is 5–8 mm, the scion can undergo safe and stable clamping. The five levels of the cutting angle corresponded to a cutting length of 4.96 mm to 6.52 mm, and the conventional cutting angle had a cutting length of 5.57 mm, all of which met the graft clip fixation requirements.

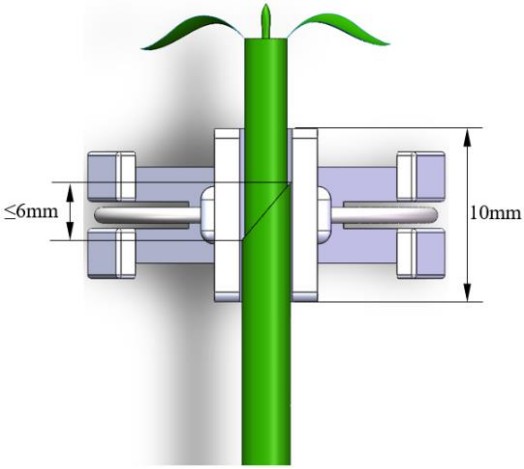

**Figure 13.** Rootstock-scion cutting butt.

### 3.3. Comparative Analysis of Cutting Quality

The statistical results of cutting quality for precision cutting and conventional cutting are shown in Figure 14.

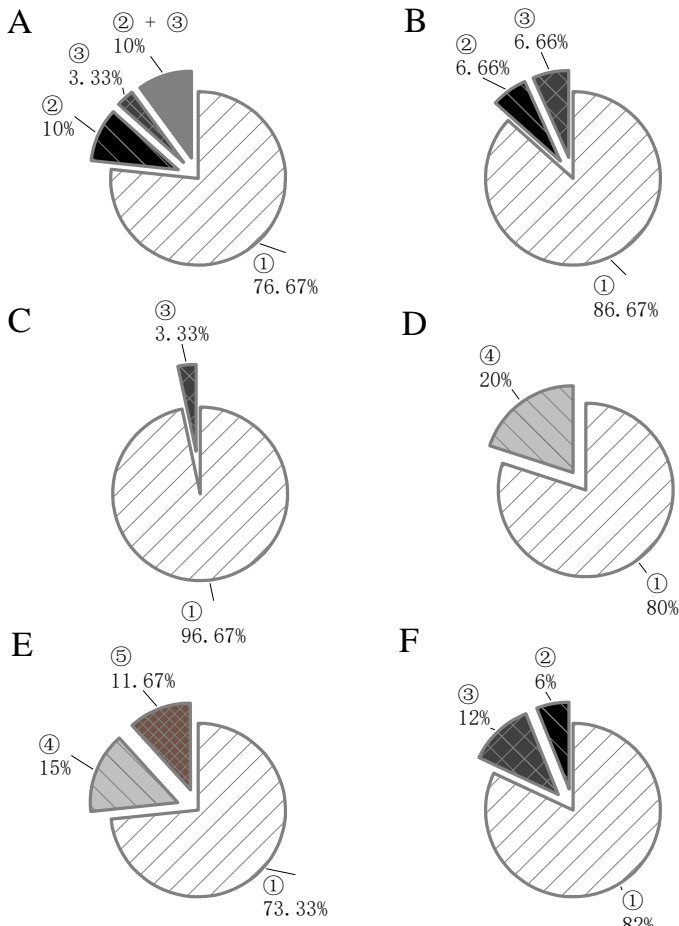

**Figure 14.** Comparison of quality of precision cutting and conventional cutting. (**A**) 22°; (**B**) 24°; (**C**) 26°; (**D**) 28°; (**E**) 30°; (**F**) Conventional cutting. ① Success rate; ② Cut through the medullary cavity; ③ Uncut clean growth point; ④ Cut cotyledons; ⑤ Cut off the leaf crown.

Cutting angles of 26°, 28°, and 30° did not result in any cutting through the medullary cavity, while cutting angles of 22° (② & ② + ③) and 24° did. The largest percentage of cutting through the medullary cavity was 20% at this angle. Similarly, the largest percentage of uncleaned growth points was 13.33% at a cutting angle of 22°, followed by 24°. The only reason for cutting failure was uncleaned growth points at 26°, and no uncleaned growth points were observed at cutting angles of 28° and 30°. The phenomenon of cutting cotyledons appeared for the first time at a cutting angle of 28° with a large percentage of 20%. However, as the cutting angle increased, the phenomenon of cutting off the leaf crown appeared at 30° with a large percentage of 11.67%. At the same time, the percentage of cutting cotyledons was still very high at 15%. The failure of conventional cutting is mainly due to cutting through the medullary cavity and failing to cut the growth point cleanly, and the failure to cut the growth point cleanly accounts for a large proportion of the possible reasons for the manual adjustment, which is not ideal for accurately grasping the cutting angle.

The difference between the cutting angle of 22° and the minimum limit cutting angle of 19.19° is small, and the cutting trajectory has a curvature inward. Thus, it is easy to cut through the medullary cavity. The rotary cutting trajectory is not linear, and the cutter trajectory has a tendency to retract when cutting the growth point, which results in the

medial growth point not being cut cleanly. As the cutting angle increases, cutting another cotyledon becomes the main cause of cutting failure. Although there is a seedling pressure mechanism to protect the reserved cotyledon, cotyledon cutting is more likely to occur as the cotyledon thickness and size of different seedlings vary and the cutting angle increases.

The results of the cutting-quality image acquisition of the rootstock are shown in Figure 15. Figure 15A,D have the best quality cuts, showing that the growth point and cotyledons were completely removed, creating a flat cut with no exposed medullary cavity. When cutting with a smaller angle (longer cut length), the concavity of the cut surface is noticeable (Figure 15B). The cut cotyledons were divided into two cases: cut petioles and cotyledons. The petiole was cut with only one layer of epidermis attached, and the cotyledon was unsupported and drooped downward in its natural state (Figure 15C), which seriously affected the healing of grafted seedlings. Cutting through the medullary cavity (Figure 15E), the scion's own roots will pass through the medullary cavity into the soil, resulting in graft failure. The growth point is not cut cleanly (Figure 15F), and the grafted seedling can survive under certain conditions. However, subsequent manual removal of the sprouting is required, increasing the workload. The irregular shape of the cut surface (Figure 15G) and cut surface epidermal pulling (Figure 15H) all affect the butt fit of the grafted seedling cuts. Cutting off the leaf crown (as in Figure 15I) means that the grafted seedling will be discarded as a whole. Therefore, cutting failure was ranked in order of impact: cut off the leaf crown, cut through the medullary cavity, cut cotyledons, and uncut the clean growth point.

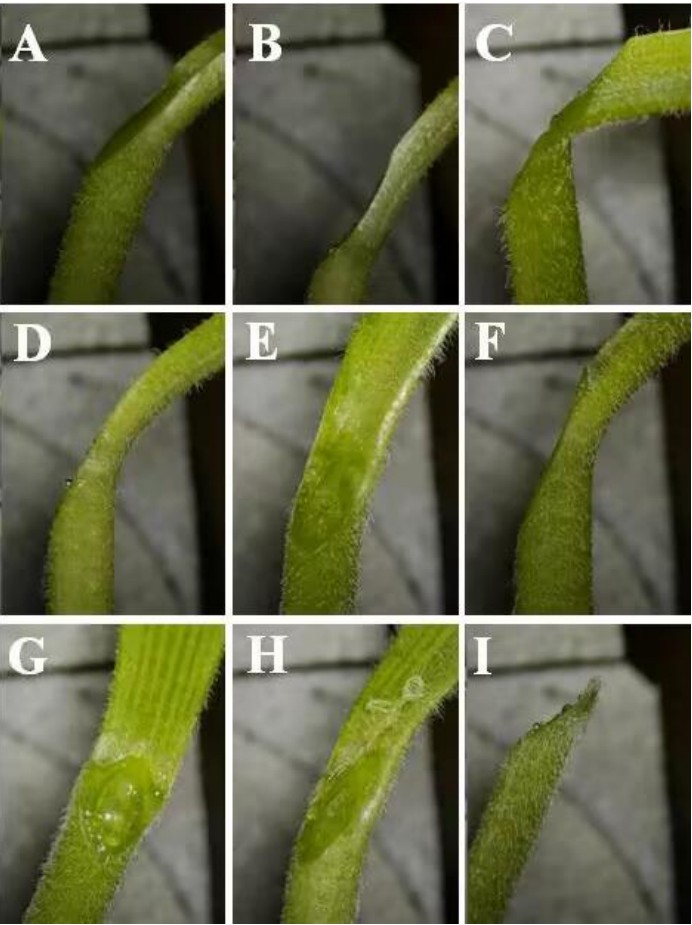

**Figure 15.** Rootstock cut surface image acquisition results. (**A,D**) Good cut surface effect. (**B**) Concavity of the cut surface. (**C**) Cut cotyledons. (**E**) Cut through the medullary cavity. (**F**) Uncut clean growth point. (**G**) Irregular shape of the cut surface. (**H**) Cut surface epidermal pulling. (**I**) Cut off the leaf crown.

In summary, a cutting angle of 22° resulted in the greatest length of the cut surface. However, the phenomena of cutting through the medullary cavity and not cutting the growth point cleanly accounted for a large proportion of the cut failure rate, and the cut surface concavity was the greatest. At a cutting angle of 24°, the phenomena of cutting through the medullary cavity and not cutting the growth point cleanly still existed, and the percentage of not cutting the growth point cleanly was higher than that at a cutting angle of 26°. A cutting angle of 30° resulted in the highest rate of cutting failure and the shortest length of the cut, which was disregarded even though the cut was the flattest. Therefore, a cutting angle of 26° is recommended to be selected as the precise cutting angle of the rootstock cutting mechanism.

### 3.4. Comparative Analysis of Cutting Success Rate

Post-hoc multiple comparisons of the five levels of cutting angles with conventional cutting are shown in Table 8. The results showed that there was no significant difference in the length of the cutting only between conventional cutting and a cutting angle of 26°, and the mean values of conventional cutting and cutting angles of 22°, 24°, 26°, 28°, and 30° differed by 0.96, 0.47, 0.06, −0.29, and −0.61, respectively. The lengths of the facets exhibited the following order: 22° > 24° > 26° > conventional cutting > 28° > 30°.

**Table 8.** Post-hoc multiple comparisons.

| (I) Independent Variable (°) | (J) Independent Variable (°) | Mean Difference (I-J) | Standard Error | Significance | 95% of Mean Confidence Interval | |
|---|---|---|---|---|---|---|
| | | | | | Lower Limit | Upper Limit |
| 22 | conventional cutting | 0.96 * | 0.06 | 0.00 | 0.84 | 1.07 |
| 24 | conventional cutting | 0.47 * | 0.06 | 0.00 | 0.36 | 0.59 |
| 26 | conventional cutting | 0.06 | 0.06 | 0.32 | −0.06 | 0.17 |
| 28 | conventional cutting | −0.29 * | 0.06 | 0.00 | −0.41 | −0.18 |
| 30 | conventional cutting | −0.61 * | 0.06 | 0.00 | −0.72 | −0.49 |

"*" The significance level for the difference in means is 0.05.

The results of descriptive statistics for conventional cutting and a cutting angle of 26° are shown in Table 9. The minimum value of cutting length for conventional cutting is 4.70 mm, and the maximum value is 6.31 mm, with a coefficient of variation of 0.07 and a mean value of 5.57 mm. The minimum value of section length for a cutting angle of 26° is 5.34 mm, the maximum value is 6.01 mm, the coefficient of variation is 0.04, and the mean value is 5.63 mm. Both of them have a similar mean value of cutting length. However, the distribution of values of conventional cutting section length is more discrete, and the distribution of values for a cutting angle of 26° is more centralized. The same conclusion can be obtained in Figure 12b.

**Table 9.** Descriptive statistics.

| Object | N/ (Seedlings) | Minimum Value of Cut Length/(mm) | Maximum Value of Cut Length/(mm) | Mean Value of Cut Length/(mm) | Standard Error of the Mean | Standard Deviation | Variance Statistic | Variation Coefficients |
|---|---|---|---|---|---|---|---|---|
| Conventional Cutting | 50 | 4.70 | 6.31 | 5.57 | 0.06 | 0.41 | 0.167 | 0.07 |
| 26° of Cutting Angle | 50 | 5.34 | 6.01 | 5.63 | 0.04 | 0.21 | 0.044 | 0.04 |

The rootstock precision cutting mechanism designed in this paper was used to carry out a comparative test of cutting at five levels of cutting angles (22°, 24°, 26°, 28°, and 30°), and the statistical results of the cutting success rate were 76.67%, 86.67%, 96.67%, 80%, and 73.33%, respectively, and the cutting angle of 26° had the highest rate of success, followed by 24°, as shown in Figure 16a. For the same batch of rootstock seedlings, there was no significant difference in cutting length between a cutting angle of 26° and conventional

cutting, but the success rate of conventional cutting was much lower than that for a cutting angle of 26°, as shown in Figure 16b.

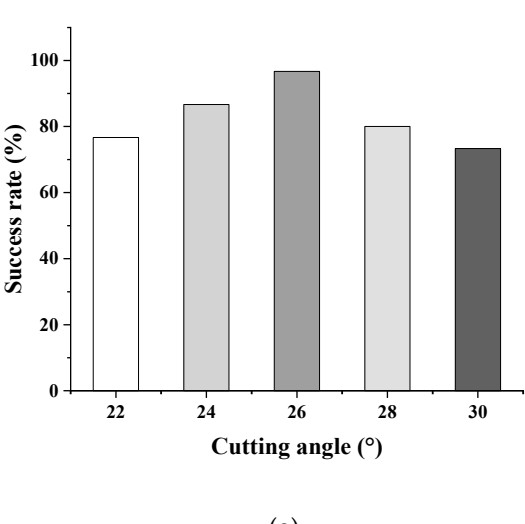

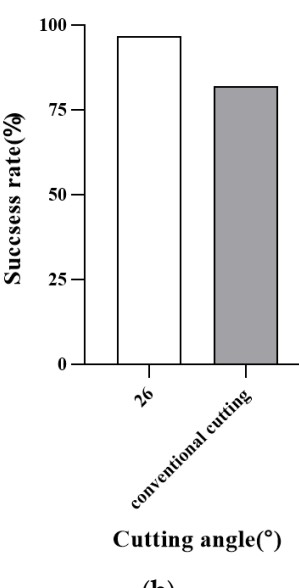

(**a**)

(**b**)

**Figure 16.** Success rate of precision cutting vs. conventional cutting. (**a**) Success rate of cutting at five levels of cutting angle; (**b**) Success rate of cutting angle 26° vs. conventional cutting.

In summary, the average cutting angle can be taken as the precision cutting angle by integrating the indexes of cutting length, cutting quality, and cutting success rate. The cutting success rate of the precision cutting angle is greater than that of conventional cutting, and the use of a precision cutting mechanism based on a motorized slide table is better than the manually adjustable precision conventional cutting mechanism.

## 4. Discussion

Due to the aging population and increasing labor shortage in China, skilled workers are in short supply during the busy season, and manual grafting is characterized by varying levels of skill, high grafting costs ($0.08/plant), and low efficiency (150–200 plants/h for the patch grafting method) [7,8]. Grafting robots can automatically graft vegetable seedlings to achieve standardization and high efficiency in nursery production, which has become an important basis for nursery enterprises to improve quality and efficiency [28]. Compared to plug-in grafting, splice grafting is very widely used in machine grafting due to its simplicity and lower requirements for seedling standardization [29]. Rootstock cutting is the core part of the grafting machine's operation process, and the success of cutting determines the subsequent healing and survival of the grafted seedlings. The traditional splice grafting method using a grafting machine adopts constant angle cutting. The machine is often no longer adjusted in the process of cutting angle parameters, and manual experience is required to judge the quality of cutting manually and adjust the cutting angle of the sliding table. In addition, the cutter distance from the position of the seedling must be manually controlled, and the cutter's up and down position needs to be adjusted many times with complex disassembly and measurements. These parameters cannot obtained the seedling's actual cutting angle. Thus, the cutting angle of the rootstock cutting for seedlings with different ages or the same age of the same seedling in different batches of seedling is difficult to determine. The cutting angle is not adaptable to different seedling ages or different batches of rootstocks of the same age.

In this study, a precise regulation method for cutting angles based on the rootstock cutting model was proposed. Modeling rootstock cutting was assessed at different growth periods to determine safe cutting areas. The motorized slide table control system was developed, and the cutter rotation center adjustment trajectory model was constructed

to convert the precise cutting angle parameters into motorized slide table displacement parameters. By comprehensively comparing and analyzing the cutting quality of five levels of cutting angle and conventional cutting in the limited cutting area (length of cut surface, quality of cut surface), the precise cutting angle of rootstock is determined to realize the safe and precise cutting of rootstock at different growth periods and to improve the success rate of rootstock cutting and the survival rate of grafted seedling healing. Exploring the precise regulation method and model of cutting angle helps the vision-guided cutting control system to realize adaptive cutting and provides theoretical prerequisites for grafting machine intelligence and automation.

Yang et al. and Tong et al. studied the fixed rotary cutting method. Most of the reasons for cutting failure were that the growth point was not cut cleanly, cutting injury to the cotyledons, and overcutting (cutting off the leaf crown), and there was no statistic on the problem of cutting failure due to cutting through the medullary cavity [21,30]. Cutting through the medullary cavity means that the grafted scion will produce adventitious roots that grow through the medullary cavity within the hypocotyl of the rootstock into the soil, losing the grafting advantage [12]. In response to the problem of grafting failure caused by scion insertion into the medullary cavity of rootstocks, Ma et al. constructed a geometric model of grafting by analyzing the internal structure of rootstocks to determine the critical point of scion insertion into the medullary cavity [31]. The premise of this study is to explore the value of the precise cutting angle of the rootstock, according to the cutting requirements of the splice grafting method (not cut through the medullary cavity, cut the growth point cleanly) to establish a cutting geometry model of the rootstock and to determine the limit cutting area of the rootstock seedling, the two limit cutting angle to form the boundary of the cutting area (19.19°, 32.53°), and the limit cutting area to the average cutting angle as the middle value, which is taken uniformly to both sides, composed of five levels of cutting angles (cannot take the boundary value, otherwise easy to cut through the medullary cavity or cut not clean growth point). The quality of cutting at five levels of cutting angle was comprehensively evaluated based on the length of the cut and the quality of the cut (cut through the medullary cavity, uncut clean growth point, cut cotyledons, and cut off the leaf crown).

Rootstock and scion are connected by vascular bundles [7,32], and the number of vascular bundles inside the seedling stem is certain. Therefore, increasing the area of the healing contact surface of the rootstock and scion can improve the healing survival rate of grafted seedlings, which is generally measured by the length of the cut surface [19]. Liang et al. analyzed the effect of different cutting angles (14°, 17°, and 27°) on the healing and survival of grafted seedlings from an agronomic point of view and found that smaller cutting angles were more favorable for grafting survival [26]. These results are similar to the results of the present study, where the length of the cut surface was longer for smaller cutting angles. The difference is that in that study, the rootstock cutting angle was empirically selected for testing. However, in this study, the 5-level cutting angle was selected based on the following criteria: determining the internal and external characteristic parameters of the rootstock in a certain growth period and calculating the data of each parameter according to the cutting model. In addition, using grafting clips to fix grafted seedlings and to provide cut stability according to the grafting clips of the clamping parameters, the length of the rootstock cut in the vertical direction shall not be greater than 8 mm. Otherwise, the scion inverted cleavage problem easily occurs, and the cut surface of the scion with a vertical height of 5–8 mm can be safely and stably clamped. According to the short axis, the diameter of seedling stems can be calculated using the upper threshold of the cut length. According to the five levels of cutting angle set in this study, the conventional cutting of the cut length is apparently in line with the requirements. The comparative analysis of the cut length with conventional cutting and five levels of cutting angles (22°, 24°, 26°, 28°, and 30°) revealed the following order: 22° > 24° > 26° > conventional cutting > 28° > 30°. This is the first result of a study comparing conventional cutting with precision cutting based on the cutting model.

Successful grafting is achieved when the difference between the rootstock and scion seedling stems is small and the angle of the cutting closely matches the contact surface [27]. Therefore, the quality of the cut surface is also an important factor in evaluating cutting success. Based on knowing the actual cutting length of five levels of cutting angle and conventional cutting and comparing the quality effect of the cutting surface, the comprehensive effect analysis seeks to select the precise cutting angle. In the author's opinion, the factors affecting cutting failure were ranked as follows: cutting off the leaf crown, cutting through the medullary cavity, cutting cotyledons, and an uncut clean growth point. The only reason for cutting failure at a cutting angle of 26° was an uncut clean growth point (3.33%); all other cutting angles and conventional cutting failure rates and reasons for failure were inferior to those obtained with a cutting angle of 26°. In addition, the cutting success rate at 26° was 96.67%, and the larger the cutting angle, the flatter the cut surface. Therefore, a cutting angle of 26° was chosen as the precise cutting angle for rootstock cutting during this period. This is the first result of a data-supported study on the selection of precise cutting angles.

*Limitations and Prospects*

(1) This paper proposes a precise cutting method for melon rootstocks, describes model construction, provides cutting parameter analysis for a specific growth period of rootstocks, and determines the precise cutting angle for each growth stage by modeling different seedling age stages.

(2) The larger the effective contact area of the rootstock and scion, the higher the survival rate of grafted seedlings, but the shape of the rootstock cut surface is irregular. There is no method for evaluating the effective butting area of the rootstock and scion, but only the rootstock seedling stem cross-section is regarded as a regular ellipse. In addition, the length of the cut is proportional to the area of the ellipse to a certain extent, so the existing method adopts the length of the cut as an index for evaluating the cut of the rootstock. The outline of the cut can be recognized and extracted using visual image technology to accurately determine the area of the cut.

(3) The cutting mechanism needs to undergo further enhancement. The cutting mechanism cutting speed needs to be further studied, and the rotary cutting power can be improved. A stepper or servo motor drive cutter rotation can be used to obtain a more suitable rotary speed for rootstock cutting, and the cutting quality and efficiency can be improved.

(4) As the theoretical premise of vision-guided adaptive cutting, the ultimate goal of this research is to construct a visual recognition and analysis system for seedling morphology, obtain the morphology information of rootstock seedlings in real time, improve the precise control of the cutting points of the rootstock, improve the precision of the cutting trajectory, and then construct a control system for closed-loop cutting to realize precise and controllable cutting.

## 5. Conclusions

In this paper, an angle-controllable precision cutting method based on a model of pith cavity cutting during the growth period of melon rootstocks is proposed. Based on the internal and external characteristics of rootstock, a precision cutting model was constructed, a precision cutting control system based on a motorized slide table was designed, the rootstock cutting mode was determined, and the rootstock precision cutting mechanism was developed. The cutting length, cutting effect, and cutting success rate of precision cutting (five levels of cutting angle in the cutting area) and conventional cutting were compared and analyzed, and the cutting quality was evaluated comprehensively. The precision cutting angle of rootstock in the period of "two leaves and one core" was selected. The conclusions of the comparison test between precision cutting and conventional cutting are as follows. The order of the length of the cutting surface is 22° > 24° > 26° > conventional cutting > 28° > 30°, and the success rates of the five levels of cutting angles (22°, 24°, 26°, 28°,

30°) and conventional cutting are 76.67%, 86.67%, 96.67%, 80%, and 73.33%, respectively. Based on comprehensive cutting length, cutting effect success rate, and other indicators, the rootstock cutting angle is determined to be 26°, which in turn provides a basis for the determination of the scion cutting angle. The research results provide the necessary technical and theoretical support for improving the rootstock-cutting accuracy of grafting robots and lay the foundation for research on and the development of a new type of rootstock precision cutting mechanism.

**Author Contributions:** Conceptualization, S.C., K.J. and Q.Z.; methodology, S.C.; validation, H.L., Q.F. and T.L.; formal analysis, L.C.; investigation, S.C., Q.Z., L.C. and K.J.; data curation, H.L., Q.F. and T.L.; writing—original draft preparation, S.C. and K.J.; writing—review and editing, Q.Z. and K.J.; supervision, L.C.; funding acquisition, K.J. All authors have read and agreed to the published version of the manuscript.

**Funding:** This research was supported by the BAAFS Innovation Ability Project (KJCX20220403), the National Nature Science Foundation of China (Grant No. 32171898), the Key R&D Project in Hubei Province (2023BBB033), and the China National Agricultural Research System (CARS-25-07).

**Institutional Review Board Statement:** Not applicable.

**Data Availability Statement:** The data presented in this study are available upon demand from the correspondence author at (jiangk@nercita.org.cn).

**Conflicts of Interest:** The authors declare no conflict of interest.

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
