# Peer review of "Melon Robotic Grafting: A Study on the Precision Cutting Mechanism and Experimental Validation"

_agriculture, doi:10.3390/agriculture13112139_

Round 1
Reviewer 1 Report
Comments and Suggestions for Authors
I have critically reviewed the manuscript entitled "Grafting Robot Rootstock Precision Cutting Mechanism, Mechanism and Experimental Validation" . Its a good work in the field of vegetable grafting as grafting method in vegetable crops especially cucurbits and solanaceous crops to get rid of various biotic and abiotic stress in a cheaper and shortest way.
I recommeneded the paper for publication.
Author Response
Dear Editors and Reviewers,
Comment: On behalf of all the authors, I would like to sincerely appreciate your valuable comments on the manuscript. Your comments not only provide constructive suggestions on improving the quality of the manuscript, but also lead us to in-depth thinking of our approaches. We will benefit from them for our future research. Based on your review comments, we have revised the manuscript accordingly and highlighted the changes. In the following, we described the changes we made corresponding to each comment.
I have critically reviewed the manuscript entitled "Grafting Robot Rootstock Precision Cutting Mechanism, Mechanism and Experimental Validation". It’s a good work in the field of vegetable grafting as grafting method in vegetable crops especially cucurbits and solanaceous crops to get rid of various biotic and abiotic stress in a cheaper and shortest way.
I recommended the paper for publication.
Authors’ Response: We gratefully appreciate for your valuable comment. Thank you for your time and dedication in reviewing our manuscript. We are honored to have had the opportunity to receive feedback from someone with your level of expertise. With your encouragement, we will work harder to send out higher level papers in the coming days.

Reviewer 2 Report
Comments and Suggestions for Authors
Although some interesting findings are presented, it is difficult to find any academic merit. I understood that the authors have developed an uniqe grafting machine, but the advanges are not shown proplery compaing to conventional machines. Authors explain the advantage of the rotary cutting method. This point is just authors’ idea, and should be proved sccientificaly at first. Or are there any previous literatures about it?
It might be very important to consider the optimizaoint on the setting angle, but it is just one process to devlop grafting machine and dose not seem to be a subject to be scientifically verified.
There are also some deficiencies in the description as a scientific papers. For example, regarding Table 1;
The values in the table might be the average values, but the number of data and the variability have to be shown at the same time. The ranges are expressed using as + -? Is it the standard deviation, maximum or minimum values, or another index? This point should also be kept in mind when creating many other tables and figures. Also, how did you define plant height and growth point height? The growth point description is given in Figure 2, but authors need to indicate where the height is measured from.
Author Response
Dear Editors and Reviewers,
On behalf of all the authors, I would like to sincerely appreciate your valuable comments on the manuscript. Your comments not only provide constructive suggestions on improving the quality of the manuscript, but also lead us to in-depth thinking of our approaches. We will benefit from them for our future research. Based on your review comments, we have revised the manuscript accordingly and highlighted the changes. In the following, we described the changes we made corresponding to each comment.
Although some interesting findings are presented, it is difficult to find any academic merit. I understood that the authors have developed an unique grafting machine, but the advantages are not shown properly comparing to conventional machines. Authors explain the advantage of the rotary cutting method. This point is just authors’ idea, and should be proved scientifically at first. Or are there any previous literatures about it?
It might be very important to consider the optimization on the setting angle, but it is just one process to develop grafting machine and does not seem to be a subject to be scientifically verified.
There are also some deficiencies in the description as a scientific papers. For example, regarding Table 1; The values in the table might be the average values, but the number of data and the variability have to be shown at the same time. The ranges are expressed using as + -? Is it the standard deviation, maximum or minimum values, or another index? This point should also be kept in mind when creating many other tables and figures. Also, how did you define plant height and growth point height? The growth point description is given in Figure 2, but authors need to indicate where the height is measured from.
Authors’ Response: We really appreciate your positive and constructive comments on our manuscript. The manuscript was revised carefully based on the comments.
- Comment: Although some interesting findings are presented, it is difficult to find any academic merit. I understood that the authors have developed an unique grafting machine, but the advantages are not shown properly comparing to conventional machines.
- Reply: As you said, we have developed the grafting machine and have been able to popularize it a lot in China. We have summarized the user feedback in detail, among which the most important one is the rootstock cutting accuracy issue, as the cutting angle adjustment of the cutting mechanism is human-operated and there is no way to know the cutting angle after adjustment. Aiming at the above problems, this paper designs the precise regulation system and realization method of the cutting mechanism by constructing the cutting model of rootstock seedlings, and derives the cutting angle suitable for melon rootstocks through experiments. Moreover, these works are significant for the improvement of the operation precision of the grafting machine.
- Comment: Authors explain the advantage of the rotary cutting method. This point is just authors’ idea, and should be proved scientifically at first. Or are there any previous literatures about it?
- Reply: Japan's ISEKI & CO., LTD. and Sangyo Research Organization, Korea's Helper Robotech Co., Ltd., China Agricultural University Professor Zhang Tiezhong, Shenyang Agricultural University Professor Tian Subo, and the author have studied the rootstock rotary cutting mechanism; many scholars have recognized the rotary cutting method. In "1 Introduction" (lines 70–78, 98–100) and "4 Discussion" (lines 618–621), these works of literature are cited, and the problems of the rootstock rotary cutting mechanism are described. Many scholars have studied and verified the effect of parameters such as cutting speed, cutter thickness, and inclination angle on the cutting quality, but the precise adjustment of cutting angle has never been solved, which also limits the use of grafting machines. The effect of parameters such as cutting speed, cutter thickness, and edge inclination angle on cutting quality, but the precise adjustment of cutting angle has never been solved, which limits the use of grafting machines.
- Comment: It might be very important to consider the optimization on the setting angle, but it is just one process to develop grafting machine and does not seem to be a subject to be scientifically verified.
- Reply: It is crucial to improve the adaptability of the grafting machine to seedlings. When the machine faces seedlings of different ages and different morphologies of rootstock seedlings, how the cutting angle changes is the key to solving the above problems. How is the rootstock cutting angle determined? It is necessary to further construct a geometric model of the internal pith cavity structure of the rootstock, determine the safe cutting area, and set different cutting angles for cutting verification, and these works provide essential prerequisites for establishing the rootstock adaptive cutting method. The angle optimization is directly related to the cutting success rate and the quality of the cut surface, which is one of the work contents that the research of rootstock cutting mechanism must determine, and the readers or users are very concerned about the cutting angle setting and optimization methods, which can also guide the actual grafting nursery production.
In addition, the internal characteristics and external morphology of rootstock seedlings of different ages vary greatly. The range of rootstock cutting maneuverability is minimal, and a slightly larger or smaller cutting angle may result in cutting failure, affecting the survival of grafted seedlings. In order to improve the cutting success rate and cutting quality, the safe cutting range of rootstock cutting seedlings should be clarified firstly. One this basis, the precise cutting angle of the cutting mechanism should be determined, and the cutting success rate and quality should be comprehensively evaluated, which needs to be scientifically analyzed and verified through the cutting test.
- Comment: There are also some deficiencies in the description as a scientific papers. For example, regarding Table 1; The values in the table might be the average values, but the number of data and the variability have to be shown at the same time. The ranges are expressed using as + -? Is it the standard deviation, maximum or minimum values, or another index? This point should also be kept in mind when creating many other tables and figures. Also, how did you define plant height and growth point height? The growth point description is given in Figure 2, but authors need to indicate where the height is measured from.
- Reply: Values in Table 1 are means; ± values are standard deviations; and the number of plants tested, 30, was supplemented in the first column of the table. After consulting the book "Experimental Design and Data Processing," the coefficient of variation is a statistic that measures the degree of variation among observations in a sample and is suitable for comparing the degree of variation in two or more samples. Table 1 is only the statistics of each appearance characteristic of the rootstock, not involved in the parameters of the comparative differences. Therefore, we think there is no need to supplement the coefficient of variation. We hope you will consider our opinion. For example, the coefficient of variation is presented in Table 9, comparing cut lengths for both cases of conventional cutting and cutting angle 26°.
Due to our negligence, the rootstocks' plant height and growth point height were not defined. Nine external characteristics, including plant height and growth point height, have been indicated in Figure 1.

Reviewer 3 Report
Comments and Suggestions for Authors
The article submitted for review is interesting because it addresses important issues for the development of agriculture, using advanced techniques to increase the technological performance of some vegetable crops.
The scientific approach is well conceived, first the theoretical aspects are discussed, then the authors elaborate the material and the research method, where the equipment designed and realized by which the grafting of different rootstocks is carried out is presented.
The results of the experimental research are then presented, confirming that the scientific approach is interesting and meets the proposed objectives.
I congratulate the authors for the results obtained, and I recommend them to continue this scientific approach in order to establish how agricultural yields are influenced.
1. The scientific approach is to design and realize a technical system through which a new method of establishing agricultural crops, respectively of grafting disease-sensitive plant varieties using resistant rootstocks, will be put into practice.
2. The article received for review addresses a problem used on a large scale for horticultural plantations and less so for the species studied by the authors of the paper.
3. The device designed in this article allows tests to be carried out for various working conditions, which, later, allow the optimization of the technological operation of grafting for the plants under study.
4. To validate the research undertaken, I recommend the authors to also consider technological aspects regarding the degree of success of the technological grafting operation, production increases, production costs, etc.
5. Considering the above, I consider that the structure and content of the article has a predominantly theoretical character and less applied.
6. The bibliographic references used could be expanded, to see other solutions made worldwide.
7. Figures 4, 10 and 11 are not clear enough.
Author Response
Dear Editors and Reviewers,
On behalf of all the authors, I would like to sincerely appreciate your valuable comments on the manuscript. Your comments not only provide constructive suggestions on improving the quality of the manuscript, but also lead us to in-depth thinking of our approaches. We will benefit from them for our future research. Based on your review comments, we have revised the manuscript accordingly and highlighted the changes. In the following, we described the changes we made corresponding to each comment.
Q1: The article submitted for review is interesting because it addresses important issues for the development of agriculture, using advanced techniques to increase the technological performance of some vegetable crops.
The scientific approach is well conceived, first the theoretical aspects are discussed, then the authors elaborate the material and the research method, where the equipment designed and realized by which the grafting of different rootstocks is carried out is presented.
The results of the experimental research are then presented, confirming that the scientific approach is interesting and meets the proposed objectives.
I congratulate the authors for the results obtained, and I recommend them to continue this scientific approach in order to establish how agricultural yields are influenced.
Q2: The scientific approach is to design and realize a technical system through which a new method of establishing agricultural crops, respectively of grafting disease-sensitive plant varieties using resistant rootstocks, will be put into practice.
The article received for review addresses a problem used on a large scale for horticultural plantations and less so for the species studied by the authors of the paper.
The device designed in this article allows tests to be carried out for various working conditions, which, later, allow the optimization of the technological operation of grafting for the plants under study.
To validate the research undertaken, I recommend the authors to also consider technological aspects regarding the degree of success of the technological grafting operation, production increases, production costs, etc.
Considering the above, I consider that the structure and content of the article has a predominantly theoretical character and less applied.
The bibliographic references used could be expanded, to see other solutions made worldwide.
Figures 4, 10 and 11 are not clear enough.
Authors’ Response: Thank you for your first comment recognizing the results of our research and I'm honored that this article has been enjoyed by you. For the second comment, we really appreciate your positive and constructive comments on our manuscript. The manuscript was revised carefully based on the comments.
- Comment: The scientific approach is to design and realize a technical system through which a new method of establishing agricultural crops, respectively of grafting disease-sensitive plant varieties using resistant rootstocks, will be put into practice.
- Reply: Your comments are invaluable, and we couldn't agree more. This paper focuses on the design of a vegetable grafting robot cutting mechanism and accurate cutting method, aiming to improve the success rate and adaptability of the grafting robot. We will follow up by inviting agronomy breeding experts to conduct further research on grafted rootstock disease-resistant varieties, cultivation of varieties suitable for mechanical grafting, etc., to form a systematic method of planting crops suitable for mechanization.
- Comment: The article received for review addresses a problem used on a large scale for horticultural plantations and less so for the species studied by the authors of the paper.
- Reply: Watermelon grafting in Beijing, China, commonly uses the Jingxin rootstock white-seeded pumpkin, and the rootstock cutting method we investigated is suitable for melon grafting rootstocks as long as the internal medullary cavity structure of the rootstock is constructed so that a suitable and precise cutting angle can be determined. Due to our carelessness, the "2.4.1 Test content" rootstock variety should be Jingxin Rootstock No. 2 white seeded pumpkin seeds. Please see line 338.
- Comment: The device designed in this article allows tests to be carried out for various working conditions, which, later, allow the optimization of the technological operation of grafting for the plants under study.
- Reply: As you said, the device needs to be tested under various operating conditions to optimize the cutting speed, cutter shape, and thickness, blade inclination angle, and other parameters to optimize the grafting technique operation. This paper tests whether the device can realize accurate cutting, a comprehensive analysis of the cutting success rate and quality. The test conditions are set as follows: cutting rotary cylinder input pressure of 0.4 MPa (converted line speed of 400 mm/s), the thickness of the cutter is 0.2mm, the tilt angle of the cutting edge is 0 °, under the above conditions can be very good to complete the cutting test. Following your comments, we will subsequently test the device under various working conditions to further optimize and improve the operation of the grafting technology.
- Comment: To validate the research undertaken, I recommend the authors to also consider technological aspects regarding the degree of success of the technological grafting operation, production increases, production costs, etc.
- Reply: We have conducted many tests and research on the efficiency, success rate, and survival rate of grafted seedlings of the grafting machine in the early stages. The grafting efficiency is 800 plants per hour, the grafting success rate is 95%, and the effect of the yield increase and the cost of production are also being tested. We will follow up with the relevant test results in an article to be announced or published. This paper aims to investigate a rootstock precision cutting mechanism, methodology, and testing and verify rootstock cutting accuracy. How to determine the precise cutting angle for rootstock seedlings of different seedling ages, in the part of the test verified the effect of five cutting angles and manual experience in adjusting the cutting angle on the cutting success rate in order to comprehensively assess the cutting effect of the cutting mechanism by the cutting success rate and the quality of the cut surface.
- Comment: Considering the above, I consider that the structure and content of the article has a predominantly theoretical character and less applied.
- Reply: The article first studied the precision cutting model of the rootstock of different seedling ages, and second, studied the cutting mechanism's precision cutting mechanism and designed the rootstock precision cutting mechanism. The test part optimizes the precision cutting angle and verifies the cutting success rate and cutting effect of the precision cutting mechanism, combining theoretical research and experimental verification with more complete content. The article's writing was inspired by the critical problems in the practical application of grafting machines, and the proposed rootstock precision cutting mechanism and precision cutting scheme provide the basis for the research of adaptive grafting robots. Subsequently, the results of this paper are integrated into a grafting machine and validated in practical applications.
- Comment: The bibliographic references used could be expanded, to see other solutions made worldwide.
- Reply: Currently, the world research on vegetable grafting machine countries is mainly Japan, Korea, China, the Netherlands, Spain, Italy, etc. European countries mainly develop grafting machines for solanaceous vegetables, and Asian countries develop grafting machines for solanaceous and melon vegetables. In "Introduction 1", the current status and problems of the latest research on cutting mechanisms for grafting machines are analyzed, and the cutting mechanisms of melon grafting machines developed by research institutes and scholars in Japan (lines 87–89), Korea (lines 89–92), China (lines 70–78, 98–105, 98–105, etc.), the Netherlands (lines 92–98), the United States (lines 112-115), and Spain (lines 115–120) are described in detail. Scholars developed cutting mechanisms for melon grafting machines and for solanaceous grafting machines in the Netherlands (lines 92–98), the USA (lines 112-115), and Spain (lines 115-120). The scope of the cited literature basically covers the global research progress on grafting machine cutting mechanisms. Therefore, we consider the reference content to be relatively complete. Please suggest more suitable references; we are more than willing to improve and supplement them.
- Comment: Figures 4, 10 and 11 are not clear enough.
7. Reply: Figure 4 is a surface rendering of the mechanism through professional software (KeyShot 11) because the material properties and colors are added according to the actual, so the figure will appear as part of the shadow, does not affect the clarity of the picture. Mechanism materials are mainly black, which is the same color as the leads, resulting in the labels and the leads may not correspond clearly, so the reviewers only need to look at the circle at the end of each lead and the corresponding number to know what it refers to; Figure 10 and Figure 11 have been replaced with higher pixel pictures instead of the original pictures. Considering the problem that the background of the pictures is not clearly distinguishable from the leads, the leads have been replaced with red color instead of black color to make them more clearly distinguishable from the background of the pictures.

Reviewer 4 Report
Comments and Suggestions for Authors
To the authors, congratulations for the excellent paper. My humble suggestion would be just to add "for the melon" at the end of the title. This is because the robot system will probably need to be adapted for other species, including other research about grafting method and cutting angle. Another aspect to be considered by the authors when including the term "for melon" concerns the possibilities of interspecific grafting, that is, if a rootstock of a species other than melon is used, the results and conclusions of this article will also be valid or not? Therefore, information on the species(s) and cultivar(s) of the scion and rootstock used must be included.
Author Response
Dear Editors and Reviewers,
On behalf of all the authors, I would like to sincerely appreciate your valuable comments on the manuscript. Your comments not only provide constructive suggestions on improving the quality of the manuscript, but also lead us to in-depth thinking of our approaches. We will benefit from them for our future research. Based on your review comments, we have revised the manuscript accordingly and highlighted the changes. In the following, we described the changes we made corresponding to each comment.
Comment: To the authors, congratulations for the excellent paper. My humble suggestion would be just to add "for the melon" at the end of the title. This is because the robot system will probably need to be adapted for other species, including other research about grafting method and cutting angle. Another aspect to be considered by the authors when including the term "for melon" concerns the possibilities of interspecific grafting, that is, if a rootstock of a species other than melon is used, the results and conclusions of this article will also be valid or not? Therefore, information on the species(s) and cultivar(s) of the scion and rootstock used must be included.
Authors’ Response: We are very grateful for your kind appraisal. Thank you for your time and dedication in reviewing our manuscript. we really appreciate your positive and constructive comments on our manuscript. The manuscript was revised carefully based on the comments.
Reply: According to the experts' opinions on the title of the article, the original title "Grafting Robot Rootstock Precision Cutting Mechanism, Mechanism and Experimental Validation" is revised to "Melon Grafting Robot Rootstock Precision Cutting Mechanism, Mechanism and Experimental Validation."
It is mentioned in "2.4.1 Test Content" (line 338) that the rootstocks were selected from white-seeded pumpkin seeds of Jingxin Rootstock No. 2. This paper focuses on precise rootstock cutting methods and, therefore, does not give information on scion varieties. The scion variety we used was Jingxin 4 watermelon seeds.

Round 2
Reviewer 2 Report
Comments and Suggestions for Authors
I think I understand the importance of this paper’s attempt. But if author has just reported the result of the optimization of machine setting, there is no scientific merit as I mentionedbefore. The objective and focus of this research have to be re-contracted in order to clarify the scientific merit of this attempt. I agree some response to reviewer. But these points are not improved in the body itself.
Author Response
Dear Editors and Reviewers,
On behalf of all the authors, I would like to sincerely appreciate your valuable comments on the manuscript. Your comments not only provide constructive suggestions on improving the quality of the manuscript, but also lead us to in-depth thinking of our approaches. We will benefit from them for our future research. Please find below the responses to the reviewers' comments.
Reviewer 2
Comment: I think I understand the importance of this paper’s attempt. But if author has just reported the result of the optimization of machine setting, there is no scientific merit as I mentioned before. The objective and focus of this research have to be re-contracted in order to clarify the scientific merit of this attempt. I agree some response to reviewer. But these points are not improved in the body itself.
Authors’ Response: Once again, thank you for your constructive comments on our article and for working with us to contribute to its quality significantly. Based on your comments, we have revised the two parts of the paper, the abstract and the introduction (the new sentences have been yellow labeled, and the rest of the changes are the switching of the positions of the sentences), which are detailed in the uploaded revised manuscript.
In the meantime, we have compiled the scientific merit of this article for your review.
The scientific merit of the study can be categorized as follows:
(1) The distribution of internal and external parameters characterizing melon rootstocks at specific growth periods was clarified. The distribution pattern of internal and external features of the rootstock was observed, and a digital representation of each feature was obtained.
(2) Areas where rootstock cutting is safe were obtained. Based on the distribution law of internal and external characteristics of rootstock, the geometric model of internal and external rootstock cutting is constructed. According to the technical requirements of the splice grafting method, the activity range of the cutter can be determined to be safe for cutting, which is different from the traditional manual cutting based on experience and realizes the precise control of the cutting angle.
(3) An automatically controlled rootstock cutting mechanism for precision cutting was designed. Different from the traditional cutting mechanism, changing the cutting angle requires tedious manual adjustment and a specific adjusted angle, and the design of the precise cutting mechanism for the human-computer interaction interface and PLC interaction of an integrated machine manually inputs the specified cutting angle the program automatically controls the motorized slide to move the corresponding displacement. It is easy to operate and can obtain a real-time cutting angle.
(4) The precise cutting angle of the rootstock cutting was determined, and the combined assessment of cutting success cut quality, and cut length was optimized. In order to make the precision cutting mechanism cut each batch of rootstock at an optimal angle, five horizontal cutting angles are selected in the safe cutting area to verify the cutting effect of the rootstock after cutting at each cutting angle. An optimal cutting angle is selected by comprehensive evaluation to improve the success rate of rootstock cutting.